# Unusual Partners: γδ-TCR-Based T Cell Therapy in Combination with Oncolytic Virus Treatment for Diffuse Midline Gliomas

**DOI:** 10.3390/ijms26052167

**Published:** 2025-02-28

**Authors:** Konstantinos Vazaios, Patricia Hernández López, Tineke Aarts-Riemens, Annet Daudeij, Vera Kemp, Rob C. Hoeben, Trudy Straetemans, Esther Hulleman, Friso G. Calkoen, Jasper van der Lugt, Jürgen Kuball

**Affiliations:** 1Princess Máxima Center for Pediatric Oncology, 3584 CS Utrecht, The Netherlands; k.vazaios@prinsesmaximacentrum.nl (K.V.); e.hulleman@prinsesmaximacentrum.nl (E.H.); f.g.j.calkoen-2@prinsesmaximacentrum.nl (F.G.C.); 2Center for Translational Immunology, University Medical Center Utrecht, Utrecht University, 3584 CX Utrecht, The Netherlands; p.hernandez-lopez@umcutrecht.nl (P.H.L.); t.aarts@umcutrecht.nl (T.A.-R.); j.t.daudeij-2@umcutrecht.nl (A.D.); g.c.m.straetemans@umcutrecht.nl (T.S.); 3Department of Cell and Chemical Biology, Leiden University Medical Center, Leiden University, 2333 ZC Leiden, The Netherlands; v.kemp@lumc.nl (V.K.); r.c.hoeben@lumc.nl (R.C.H.); 4Department of Hematology, University Medical Center Utrecht, 3584 CX Utrecht, The Netherlands

**Keywords:** diffuse midline glioma, oncolytic viruses, immune-oncology, immunotherapy, γ9δ2TCR, TEGs, D24-RGD, R124

## Abstract

Due to the minimal survival benefits of existing therapies for pediatric diffuse midline glioma (DMG) patients, new therapeutic modalities are being investigated. Immunotherapies such as CAR-T cells and oncolytic viruses (OVs) are part of these efforts, as evidenced by the increasing number of clinical trials. αβ T cells engineered with a high-affinity γ9δ2 T-cell receptor (TEGs) are immune cells designed to target metabolic changes in malignant or virally infected cells via BTN2A1 and BTN3A. Because the expression of BTN2A1 and BTN3A can be altered in tumor and infected cells, combining TEGs and OVs could potentially enhance the anti-tumor response. We investigated this hypothesis in the following study. We demonstrate that TEGs can indeed target DMG, which expresses BTN2A1 and BTN3A at varying levels, and that OVs can further enhance the expression of BTN3A—but not BTN2A1—in DMG. Functionally, TEGs killed DMG cell cultures, and this killing was further increased after OV infection of the DMGs with either adenovirus Δ24-RGD or reovirus R124 under suboptimal conditions. However, this additive effect was lost when γ9δ2 TCR–ligand interaction was boosted by pamidronate. This study demonstrates the additive effect of combining OVs and Vγ9Vδ2 TCR-engineered immune cells under suboptimal conditions and supports a combination strategy to enhance the efficacy of both therapeutic modalities.

## 1. Introduction

Immunotherapies such as anti-GD2 CAR-T cell therapies are being evaluated in clinical trials for pediatric diffuse midline gliomas (DMGs) as potential therapeutic agents [1]. While CAR-T cell therapy is promising in liquid tumors, the frequently high heterogeneity of tumor antigen expression and anti-inflammatory microenvironment limit their success in DMG [2,3]. Therefore, alternative therapeutic agents are urgently needed, preferably ones that have additive or synergistic effects with other therapies. γδ T cells and, in particular, γ9δ2T cells have been reported to have strong anti-tumor reactivity. A major increase in the understanding of the ligands of γ9δ2T cells has enabled studying therapeutic strategies for γ9δ2T cell receptor-based therapies in further detail. Their mode of action includes the upregulation of the butyrophilin 2A1 (BTN2A1) [4] which forms a complex with BTN3A1/2 heterodimers through phosphoantigens [5]. This process is heavily regulated by the small GTPase RhoB [6] and depends on the trafficking of BTN3A1 [7]. γ9δ2TCR detection acts on metabolic stress in cancer cells and depends on AMP-activated protein kinase (AMPK) [8]. Based on these molecular insights, our group developed a novel therapeutic named TEGs (αβ T cells engineered with a high-affinity γ9δ2TCR) [9,10]. A first clinical remission has been observed in an ongoing Phase I clinical trial in acute myeloid leukemia and multiple myeloma. In addition, many years of investigations have demonstrated the role of γδ T cells in recognizing infected cells [11]. In particular, the role of γ9δ2T cells has been reported in recognizing Epstein-Barr virus (EBV)-infected cells [12], while other γδ T cell subsets recognize cells that are infected with cytomegalovirus (CMV) [13]. Although the responses of human γδ T cells and their antiviral abilities following allogeneic hematopoietic stem cell transplantation have been extensively examined in infections with herpes viruses like CMV or EBV, there is a lack of studies addressing the involvement of γδ T cells in other herpes viruses such as varicella-zoster virus (VZV), as well as non-herpes viruses like adenoviruses (ADV) and reoviruses (RV) [14]. Thus, TEGs might be able to elicit both anti-tumor and antiviral responses.

Clinical trials with oncolytic viruses (OV), such as the ADV DNX-2401, are among the few clinical trials that have demonstrated improved progression-free survival, extending from the historical 12 months to 18 months [15]. OVs infect tumor cells and kill them during their infection cycle. Thus, creating new viral particles that repeat the process by infection of nearby uninfected tumor cells can lead to immunogenic cell death (ICD) and induction of an anti-tumor and antiviral response [16,17,18]. DNX-2401, also preclinically known as D24-RGD, is a type 5 human adenovirus genetically modified to target cells expressing alpha-V integrins and replicating in cycling cells and cells with abnormalities in their retinoblastoma (RB) pathway, providing tumor specificity [15,19]. R124 is an unmodified human reovirus T3D that has been reported to have tropism towards cycling cells and cells with an upregulated Ras/RalGEF/p53 pathway [20]. Both OVs have been investigated in vitro in a number of brain tumors, including pediatric DMGs, demonstrating their variable effect on different DMG cell cultures [21,22,23].

In summary, based on the intrinsic targeting characteristics of TEGs that allow them to target infected cells and metabolically aberrant cells, we expected an additive or synergistic effect between OVs and TEGs for tumor clearance. Therefore, we aimed to explore the possibility of TEG and OV combination therapy of pediatric brain tumors (PBTs), using an in vitro model of patient-derived DMG cell cultures to follow the cytotoxic action of both modalities. To accomplish this, two DMG cell cultures were used and infected with two major OVs currently explored in clinical practice: ADV (D24-RGD) and RV (R124)-based OVs.

## 2. Results

### 2.1. TEGs Recognize Pediatric DMGs Through Their γ9δ2TCR

First, we investigated the ability of TEGs to induce a cytotoxic effect against SU-DIPG-IV and VUMC-DIPG-G, two patient-derived DMG cultures expressing different levels of membranous BTN3A and BTN2A1, required for γ9δ2TCR interaction with tumors (Figure 1A). Therefore, DMG cells were co-cultured with TEGs and LM-1 cells (non-functional TEGs) either in the absence or presence of 30μM pamidronate (PAM) for 48 h. At the end of the 48 h co-culture in the presence of PAM, TEGs were able to significantly induce the killing of both DMGs, as observed by the reduced number of tumor cells (Figure 1B). In addition, cytokine secretion of TEGs also significantly increased in the presence of PAM, further confirming the ability of TEGs to target DMGs (Figure 1C). Thus, we conclude that DMGs are susceptible to TEG in the presence of PAM.

### 2.2. OVs Kill DMGs and Enhance BTN3A but Not BTN2A1 Expression

Next, we assessed the susceptibility of DMG cells for ADV D24-RGD and RV R124 after 48-h infection at several multiplexity of infection (MOI) from 5 to 50 for D24-RGD and 20 to 300 for R124. Both D24-RGD and R124 induced killing of SU-DIPG-IV and VUMC-DIPG-G in a dose-dependent manner ranging from 10% to 60% (D24-RGD) and 15% to 40% (R124) (Figure 2A,B). We selected the MOI 5 for D24-RGD and 20 for R124, which induced less than 20% cytotoxicity, to investigate their combination potential with TEGs. To test whether OVs could have a potential synergistic effect with TEGs, we tested the impact of OVs on the expression of components of the multimeric ligand complex of the Vγ9Vδ2 TCR-engineered TEGs, namely BTN2A1 and BTN3A. Expression of BTN2A1 and BTN3A in SU-DIPG-IV and VUMC-DIPG-G was investigated at 2 days post-infection with D24-RGD and R124 (Figure 2C,D). BTN2A1 expression, the molecule that is needed for the first binding step of the γ9δ2TCR, remained unchanged for both cell cultures (Figure 2C). In contrast, BTN3A was significantly upregulated after infection with 20 MOI R124 in SU-DIPG-IV and VUMC-DIPG-G, while 5 MOI D24-RGD significantly upregulated BTN3A in VUMC-DIPG-G (Figure 2D). This upregulation was also observed on infected cells in the presence of 30μM PAM (Figure 2C,D). These data supported further investigation for beneficial combinations of TEGs and OVs.

### 2.3. Additive Effect of the Combination of OVs and TEGs to Enhanced DMG Killing

After confirming that both TEGs and OVs induce cytotoxicity in both DMG cell cultures tested and that OVs enhance BTN3A expression, we combined these therapeutic modalities to explore the combined net effect. In the presence of 30µM PAM, an optimized ligand expression for the γ9δ2 TCR, TEGs demonstrated an enhanced activity against SU-DIPG-IV and VUMC-DIPG-G. However, the addition of D24-RGD or R124 did not further increase the killing efficacy of the TEGs (Figure 2E,F). In contrast, a decrease in the percentage of hexon (D24-RGD) positive or σ3 (R124) positive DMGs was observed in the presence of 30µM PAM, especially in higher concentrations (Appendix A). Interestingly, in the absence of PAM, when the γ9δ2TCR interaction with its ligands BTN2A1 and BTN3A is suboptimal, the combination of D24-RGD with TEGs resulted in a significant increase in tumor killing from approximately 13% to 30% for SU-DIPG-IV cells (Figure 2G), whereas the combination with R124 increased the killing of VUMC-DIPG-G cells from 16% to 40% (Figure 2H). When the combinations of OVs and TEGs were tested in higher MOI, the OV monotherapy reached similar levels of cytotoxicity compared to the combination with the TEGs and OVs (Appendix A); thus, removing the TEG-specific combination benefit.

As we observed enhanced killing in suboptimal conditions for TEGs and OVs and increased membranous BTN3A levels through OV infection, we tested whether this effect improved TEG activation. Therefore, we used IFN-γ release as a readout to assess TEG activation by the OVs; if OVs increased TEG activation, we would expect an impact on IFN-γ secretion mediated by TEGs. In all combinations with TEGs, no statistically significant IFN-γ increase was noted when TEGs were co-cultured with D24-RGD- or R124-infected DMGs, either in the absence or presence of PAM (Figure 2I,J). While a small yet insignificant increase was noted with D24-RGD infection of DMGs, a small IFN-γ decrease in release was demonstrated with R124 infection in the presence of PAM (Figure 2I,J). These results imply that, despite the impact of OVs on BTN3A surface expression, the observed benefit of the combination therapy is most likely mainly additive and not induced by additional activation of the TEGs by the presence of the OVs.

## 3. Discussion

With the current rise of immunotherapies, several modalities are being investigated for the treatment of PBTs, including DMG [16,24]. CAR-T cells are a great example of such therapies, demonstrating promising results against many tumor types. However, in some cases, the initial tumor regression after CAR-T infusion was followed by tumor antigen escape that led to tumor progression [25]. Within this context, we show that targeting BTN2A1 and BTN3A through TEGs might add to the arsenal of cellular cancer immune therapies for DMG. In our experiments, recognition of DMGs, however, depended on PAM, implying the importance of bisphosphonates for TEG-based therapies, as also demonstrated in an ongoing clinical trial [6,10,11,26]. We hypothesized that viral infection might enhance γ9δ2TCR efficacy since the γ9δ2TCR recognizes oncogenic transformations that highly overlap with the transformations occurring in infected cells [27]. The selected OVs, D24-RGD and R124, upregulated BTN3A, a part of the ligand complex for γ9δ2TCR [5]. In contrast, the counterpart of BTN3A, namely BTN2A1, which is required for successful interaction with the γ9 chain of the γ9δ2TCR [4], was not increased on the cell surface after OV infection. Importantly, successful combination therapy of TEGs with OVs was not observed when BTN3A expression increased, and the effect was additive rather than synergistic. Thus, the optimal BTN2A1/BTN3A complex formation induced by PAM might be more important than the total expression of both proteins [5].

The molecular mechanism underlying why only BTN3A was affected by OV remains unclear. However, recent studies detailing the differential regulation of BTN2A1 and BTN3A may provide valuable insights to help explain our observations [8,26]. More specifically, the induction of both BTN3A and BTN2A1 partially depends on AMP-activated protein kinase (AMPK) [8]. In addition, single oncogenic mutations that activate the phosphoinositide 3-kinase (PI3K)/AKT1 pathway have been reported as a prerequisite of BTN2A1 upregulation during early oncogenic transformation [26]. BTN3A upregulation on cancer cells has been reported to be partially PAM-dependent, where phosphorylation of BTN3A, RHOB, PHLDB2, SYNJ2, and CARMIL1 have been shown to be mainly involved in orchestrating PAM-induced cytoskeletal rearrangements in tumor cells required for spatial and conformational changes in BTN3A [6,26]. On the other hand, adenoviruses such as D24-RGD have been shown to upregulate host proteins that activate the PI3K/mTOR pathway in order to facilitate viral replication [28,29]. Thus, OVs might be influencing the latter pathway even without the need for additional PAM treatment.

Regardless of the underlying molecular mechanism of BTN3A upregulation, our data suggest that an additional increase in BTN3A expression mediated solely by OVs does not further augment TEG activity, as lack of BTN2A1 upregulation or an optimized complex formation was not supported after administration of OVs. The lack of a synergistic mechanism was further steered by the absence of an increase in IFN-γ release in co-cultures with virally infected DMGs. Since also a small decrease in IFN-γ release was observed in co-cultures of R124-infected DMGs in PAM, this suggests incompatibility of PAM and OVs. In addition, PAM reduced the number of D24RGD- and R124-infected cells as we observed (Appendix A), which could be attributed to the metabolic altering properties of PAM, such as reduction of Ras, necessary for R124 replication or by reduction of membrane motility affecting viral entry [30,31]. Interestingly, Zheng et al. previously demonstrated the treatment of mice infected with lethal avian influenza H7N9 virus using PAM as an antiviral agent [32]. However, as we used IFN-γ secretion by TEGs as the sole readout for potential synergy, we may have underestimated the additive functional effect of TEGs resulting from OV–Tumor infection.

Despite the limited combinational capacity of TEGs and OVs in the presence of PAM, improved DMG killing was observed when TEGs were used without PAM and when OVs were applied at lower concentrations. This finding could be of clinical relevance, as TEGs may encounter suboptimal ligand expression and/or limited infiltration of PAM in the tumor due to pharmacokinetics, as PAM has been demonstrated to be mainly absorbed into bones in adult and pediatric patients and other soft tissues [33,34], while OVs, due to their topical application and tumor tropism, may benefit the TEG action against DMGs when PAM presence is depleted in the tumor area, leading to an OV-induced long-term activation of TEGs.

For successful implementation of OV therapy, OVs should demonstrate (a) high tropism towards the tumor cells, (b) high replication and infection rate, (c) low immunogenicity to avoid antiviral response, and (d) high activation of the immune response [35]. The tumor tropism of the OVs greatly differs by virus and is usually not restricted by the tumor type but is dependent on intrinsic factors such as the tumor cell metabolism [23,36], affecting their oncolytic efficacy. The benefit of OVs over other immune therapies is their ability to persist in the immunosuppressive microenvironment induced by the tumor and their capacity to evade the “immune-editing” of the tumors, while their abilities to (a) debulk the tumor and (b) activate the immune microenvironment through the ICD make them perfect candidates for combination with other immunotherapies [16]. The heterogeneity observed in OV responses could also affect the OV-induced immune activation, as previously demonstrated in glioblastoma patient-derived co-cultures [37]. This immune-regulatory heterogeneity could also explain the successful combination of D24-RGD with TEGs only observed against SU-DIPG-IV and of R124 with TEGs only observed against VUMC-DIPG-G.

This study acts as a proof of concept for the combination of OVs with TEGs, and further optimization might be required to increase the rate of successful combinations. Factors such as OV and TEG concentrations, as well as the timing of OV and TEG administration, might be key factors of success [38]. In addition, other cell populations present in the tumor microenvironment might be required for the successful implementation of this strategy. On the one hand, in vitro models at the moment are unable to recapitulate these complex interactions, and on the other hand, immunocompetent mice cannot reproduce the oncolytic effects of the OVs due to species tropism as human viruses usually replicate poorly or cannot infect other organisms, thus requiring the use of other less common in vivo platforms [39]. Consequently, as many animal studies do not fully translate from bench to bedside, patient-derived cell cultures—though containing less cellular context—are being used for proof of concept studies and the identification of therapeutic targets or the investigation of direct therapy-tumor interactions [40,41].

In conclusion, we demonstrated that TEGs in the absence of PAM and low concentrations of OVs have an additive cytotoxic effect on DMG cell cultures. This indicates that the combination of TEG and OV therapy is a promising novel therapeutic strategy for DMG, as well as for other solid tumors with few known tumor-specific antigens.

## 4. Materials and Methods

### 4.1. DMG Cell Cultures

SU-DIPG-IV were provided by Dr. Monje (Stanford University, Stanford, CA, USA) [42] while VUMC-DIPG-G were established from autopsy material at the Amsterdam UMC (Vrije University Medical Center of Amsterdam, Amsterdam, The Netherlands). The cells were cultured at 37 °C and 5% CO_2_ in Tumor Stem Medium (TSM) consisting of 48% Neurobasal-A medium (#10888022, Thermo Fisher, Amsterdam, The Netherlands), 48% DMEM/F12 with Phenol Red without glutamine (#31330095, Thermo Fisher, Amsterdam, The Netherlands), 1% HEPES 1M (#15630-080, Thermo Fisher, Utrecht, The Netherlands), 1% MEM Non-essential amino acid solution (#11140050, Thermo Fisher, Utrecht, The Netherlands), 1% Sodium pyruvate 100 mM (#11360039, Thermo Fisher, Utrecht, The Netherlands), and 1% Glutamax (#35050038, Thermo Fisher, Utrecht, The Netherlands) (TSM base). TSM base was supplemented with 2% B27 without vitamin A (#12587010, Thermo Fisher, Utrecht, The Netherlands), 20 ng/mL human Epidermal Growth Factor (#AF-100-18B-1MG, Peprotech, Cranbury, NJ, USA), 20 ng/mL human Basic Fibroblast Growth Factor (#AF-100-18B-1MG, Peprotech, Cranbury, NJ, USA), 10 ng/mL human Platelet-derived Growth Factor AA (#100-13A-250uG, Peprotech, Cranbury, NJ, USA), 10 ng/mL human Platelet-derived Growth Factor BB (#100-14B-250uG, Peprotech, Cranbury, NJ, USA), and 5000 U/mL Heparin and 1% penicillin/streptomycin (P0781-100ML, Sigma Aldrich, Amsterdam, The Netherlands) (Complete TSM).

### 4.2. T Cell Transduction to Express a γδ TCR (TEGs and LM1s)

TEGs (T cells engineered to express a highly tumor-reactive Vγ9Vδ2 TCR) and LM1s (mock T cells engineered to express a mutant γ9δ2TCR with abrogated function) and Jurkat-76-γδTCR (Jurkat-76 cells engineered to express a highly tumor-reactive Vγ9Vδ2 TCR) were expanded and selected as previously described [43]. Briefly, helper constructs gag-pol (pHIT60), env (pCOLT-GALV), and pMP71 retroviral vectors containing both γ9δ2TCR chains separated by a ribosomal-skipping T2A sequence were transfected in packaging cells (Phoenix-Ampho) using FugeneHD reagent (Promega, Leiden, The Netherlands). Phoenix-Ampho were then cultured in DMEM + GlutaMAX (#10569010, Gibco, Utrecht, The Netherlands), 10% heat-inactivated FCS (Sigma Aldrich, F0804, Amsterdam, The Netherlands), and 1% penicillin/streptomycin for at least 24 h at 37 °C and 5% CO_2_. Healthy donors derived human peripheral blood mononuclear cells (PBMCs) were preactivated with anti-CD3 (30 ng/mL, Orthoclone OKT3, Janssen-Cilag, Leiden, The Netherlands) and IL-2 (50 IU/mL, Proleukin, Novartis, Amsterdam, The Netherlands) and cultured in 1640 RPMI+ Glutamax (#61870036, Gibco, Utrecht, The Netherlands) supplemented with 5% pooled and heat-inactivated human serum (Sanquin, Utrecht, The Netherlands), 0.5 M beta-2-mercaptoethanol, and 1% penicillin/streptomycin (huRPMI). The supernatant collected from the Phoenix-Ampho containing retrovirus was filtered with a 0.45 µm strain, and the preactivated PBMCs or Jurkat-76 cells were subsequently transduced twice with viral supernatant within 48 h in the presence of 6 mg/mL polybrene (Sigma-Aldrich, Amsterdam, The Netherlands). For PBMCs, 50 IU/mL IL-2 was also added during transduction. After transduction, the TCR-transduced T cells were expanded by stimulation with anti-CD3/CD28 Dynabeads (500,000 beads 10–6 cells; Life Technologies, Bleiswijk, The Netherlands) and IL-2 (50 IU/mL). During expansion, half the huRPMI was removed and renewed with new fresh containing only IL-2 (50 IU/mL) every 3–4 days. After a week of expansion, the Vγ9/Vδ2TCR-transduced T cells were depleted of non-engineered T cells by magnetic-activated cell sorting (MACS) using a biotin-labeled anti-αβ TCR (clone BW242/412; Miltenyi Biotec, Leiden, The Netherlands) and subsequently incubated with anti-biotin coupled to magnetic beads (anti-biotin MicroBeads, 130-090-485, Miltenyi Biotec, Leiden, The Netherlands). Next, labeled cells were loaded onto an LD column, and αβ T cells were depleted by MACS following the manufacturer’s protocol. After depletion, TEGs were expanded using a T cell rapid expansion protocol 14. Jurkat-76- γδTCR cells were cultured in RPMI with 1% penicillin/streptomycin and 10% heat-inactivated FCS, and selected for CD3 using CD3 MicroBeads (130-097-043, Miltenyi Biotec, Leiden, The Netherlands) following manufacturer’s protocol.

After γ/δTCR+ TEGs and LM1s were stimulated for 2 weeks in huRPMI on a feeder cell mixture comprising sublethally irradiated allogenic PBMCs, Daudi and LCL-TM in the presence of IL-2 (50 IU/mL), IL-15 (5 ng/mL; both R&D Systems, Minneapolis, MN, USA), and PHA-L (1 μg/mL; Sigma-Aldrich, Amsterdam, The Netherlands) according to the rapid expansion protocol (REP) with refreshment of the huRPMI supplemented with IL-2 and IL-15 [43]. To monitor the purity of γ/δTCR+, as well as the absence of allogenic irradiated feeder PBMCs, cells were analyzed weekly by flow cytometry using the antibodies anti-pan γδTCR-PE (IM1418U, Beckman Coulter, Woerden, The Netherlands), anti-αβTCR-FITC (17-9986-42, eBioscience, San Diego, CA, USA), anti-CD8-PerCP-Cy5.5 (301031, Biolegend, Amsterdam, The Netherlands), and anti-CD4-APC (300514, Biolegend, Amsterdam, The Netherlands). TEGs of purity < 90% were re-isolated as described above to increase purity.

After at least one REP cycle of expansion, the TEGs and LM-1 underwent subsequent CD4 and CD8 positive isolation using either CD4 or CD8 Microbeads (130-097-048 & 130-045-201, Miltenyi Biotech, Leiden, The Netherlands) following the manufacturer’s instructions. After incubation with magnetic microbeads, cells were applied to LS columns, and CD4+ or CD8+ TEGs or LM1s were selected by MACS. After the MACS selection procedure, γ9δ2TCR+ CD4+ or γ9δ2TCR+ CD8+ subsets of TEGs were stimulated every 2 weeks using REP. To monitor their purity, the antibody panel mentioned before was used and the cells were analyzed by flow cytometry weekly. CD8+ and CD4+ TEGs or LM-1 of >90% purity were used for functional assays. TEGs or LM-1 were used for co-culture assays 4–5 days after the last IL2/IL15/PHA-L stimulation.

### 4.3. Co-Culture Functional Assays

DMG cells were collected, washed in cold PBS, and dissociated into single cells with accutase (A6964-100ML, Merck, Amsterdam, The Netherlands), washed in cold PBS, and stained with 5 μM of Cell trace violet (CTV) (#10220455, Invitrogen, Utrecht, The Netherlands) for 20 min at 37 °C and 5% CO_2_. After CTV staining, CTV was neutralized in media with FCS for 2 min and then washed away by PBS. The cells were counted using trypan blue 0.4% and put in TSM. A total of 25.000 tumor cells/well were seeded in 96-well plates and incubated for 1 h in TSM at 37 °C and 5% CO_2_. After incubation, DMGs were infected with 5, 10, 20, 50 MOI of D24-RGD or 20, 100, 300 MOI of R124 in RPMI + 10% FCS and 25.000 cells/well. TEG/LM-1 in RPMI + 10% FCS was also added where needed, resulting in a 50/50 concentration of TSM and RPMI in all conditions with/without 30 μM pamidronate (PAM). At 48 h post-infection, flow-count Fluorospheres (7547053, Beckman Coulter, Woerden, The Netherlands) were added to the wells for normalization purposes, and then the plates were centrifuged. After centrifugation, the supernatant was collected to be used for IFN-γ release assays. The cell pellet was treated with accutase for 5 min with frequent shaking of the plates for sphere and cell cluster dissociation into single cells. After dissociation, the wells were washed with FACs buffer (PBS + 1%BSA + 0.05% sodium azide) and stained with anti-CD3 -AF700 (#300424, Biolegend, Amsterdam, The Netherlands) and 7-AAD-PE-Cy5 (#130-111-568, Miltenyi Biotech, Leiden, The Netherlands) for 30 min at 4 °C. After successful staining, the wells were washed with FACs buffer and fixed in 4%PFA in PBS for at least 20 min at 4 °C. After fixation, the plates were analyzed with flow cytometry using the Fortezza (BD biosciences, Drachten, The Netherlands), FACs gating to calculate the alive tumor cells (CTV+ and 7-AAD- cells) and analyses were processed in BD FACSDIVA (BD biosciences, Drachten, The Netherlands).

### 4.4. IFN-γ Release Assay

IFN-γ release was calculated using the Ready-set-go human IFN-γ ELISA (# 88-7316-22, Thermo Fisher, Utrecht, The Netherlands) from the collected supernatants. Briefly, MaxiSorp™ 96-Well Plates (Polystyrene) Clear were coated overnight at 4 °C with capture antibody in coating buffer. After coating, the plates were washed five times with wash buffer (PBS + 0.5% Tween 20), and then the plates were incubated for one hour with assay diluent in RT, followed by five washes. Next, the supernatant was incubated for two hours, with standard and serial 2-fold dilutions of the standard for the creation of the standard curve, followed by five washes. The detection antibody was added to the wells and incubated in RT for one hour, and after five washes, avidin-HRP enzyme was then added for a half-hour incubation in RT and washed five times. Finally, tetramethylbenzidine (TMB) substrate solution was added to each well for 15 min to react with the HRP at RT, and then sulfuric acid was added as a stop solution to each well. The plates were in a plate reader at 450 nm and 570 nm. The values of 570 nm were subtracted from the 450 nm, and the IFN-γ concentrations were determined through the values of the standard curve.

### 4.5. BTN2A1 and BTN3A Staining

DMG cells were collected, washed in cold PBS, and dissociated into single cells with accutase for 5 min at 37 °C and 5% CO_2_. The cells were counted using trypan blue 0.4%, and 20,000 tumor cells/well were seeded in 96 wells in TSM. After 1 h incubation, the DMG cells were infected with 5 MOI of D24-RGD or 20 MOI R124 or 30 μM PAM for 48 h. At 48 h post-infection, the DMG cells were centrifuged, the supernatant was removed, and the cell pellet was washed in cold PBS and treated with accutase for 5 min at 37 °C and 5% CO_2_. After dissociation, the cells were washed with FACs buffer and stained with live/dead^TM^ fixable aqua (#L34957, ThermoFisher, Utrecht, The Netherlands), anti-BTN3A1/2/3-PE (#FAB7136P-025, R&D systems, Minneapolis, MN, USA), and anti-BTN2A1-Alexa Fluor^TM^ 647 for 30 min at 4 °C. The anti-BTN2A1 antibody originated from synthetic gene fragments (Twist Biosciences, San Fransisco, CA, USA) of the patent WO/2019/057933 containing the VH and VL chains of anti-BTN2A clone 5.28, subcloned into pEE14.4-IgG1 and pEE14.4-κLC vectors and transfected into HEK293F with pAdVAntage (accession no. U47294; Promega, Leiden, The Netherlands) in a 1:2:2 ratio, using Polyethylenimine (PEI) (#19850-100, Polysciences, Warrington, PA, USA) as transfection reagent. Anti-BTN2A1 was purified from the supernatant by affinity chromatography using HiTrap™ rProtein A FF column (#17-5079-01, VWR, Amsterdam, The Netherlands). Purified Anti-BTN2A antibody was labeled using Alexa Fluor™ 647 Antibody Labeling Kits (#A20186, ThermoFisher, Utrecht, The Netherlands). After staining, the cells were washed and fixed in 4% PFA for at least 20 min at 4 °C. Finally, the plates containing the fixated cells were analyzed with flow cytometry using the Cytoflex (BD biosciences, Drachten, The Netherlands); FACs gating allowed the measurement of BTN2A1 and BTN3A median fluorescent intensity (MFI) while excluding the dead cells, analyses was processed in FlowJo (BD biosciences, Drachten, The Netherlands).

### 4.6. Plots and Statistical Analyses

Graphs were plotted and statistical analyses were calculated using GraphPad Prism 9. One-way ANOVA corrected for multiple comparison with Tukey’s test was used to compare the BTN2A1, BTN3A and infection rate changes across the conditions as well as the killing capabilities of both monotherapies and both combinations. To compare IFN-γ release, the two-tailed *t*-test was used.

## Figures and Tables

**Figure 1 ijms-26-02167-f001:**
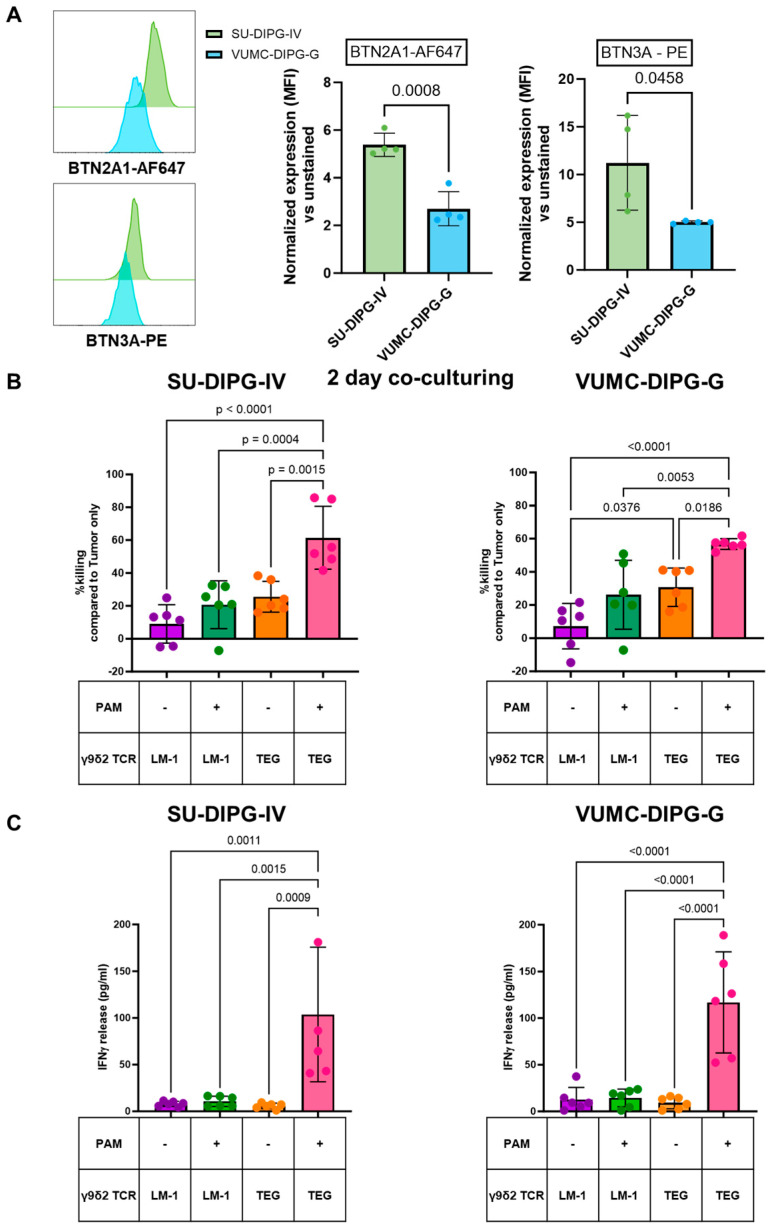
Patient-derived diffuse midline glioma cell cultures expressing BTN2A1 and BTN3A are targeted by TEGs. (**A**) BTN2A1 and BTN3A expression of SU-DIPG-IV and VUMC-DIPG-G, as normalized median fluorescent intensity (MFI) vs. their unstained counterparts. (**B**) The 48-h co-culture killing assays of SU-DIPG-IV and VUMC-DIPG-G with LM-1 and TEG cells in the absence and presence of 30μM PAM, (**C**) INFγ release after 48 h of SU-DIPG-IV and VUMC-DIPG-G with LM-1 and TEG cells in the absence and presence of 30 μM PAM. Data from multiple independent experiments are shown for BTN2A1 and BTN3A expression (n = 2) and for the killing and IFN-γ experiments (n = 3). Biological replicates are represented as dot points and the error bar as ±SD. BTN2A1 and BTN3A data were normalized on the MFI of unstained SU-DIPG-IV and VUMC-DIPG-G, respectively. Statistics were assessed with the Student’s two-tailed *t*-test. Killing data are normalized on the number of alive tumor cells without any treatment (tumor only). Statistics were assessed with one-way ANOVA and corrected for multiple comparisons with the Tukey test.

**Figure 2 ijms-26-02167-f002:**
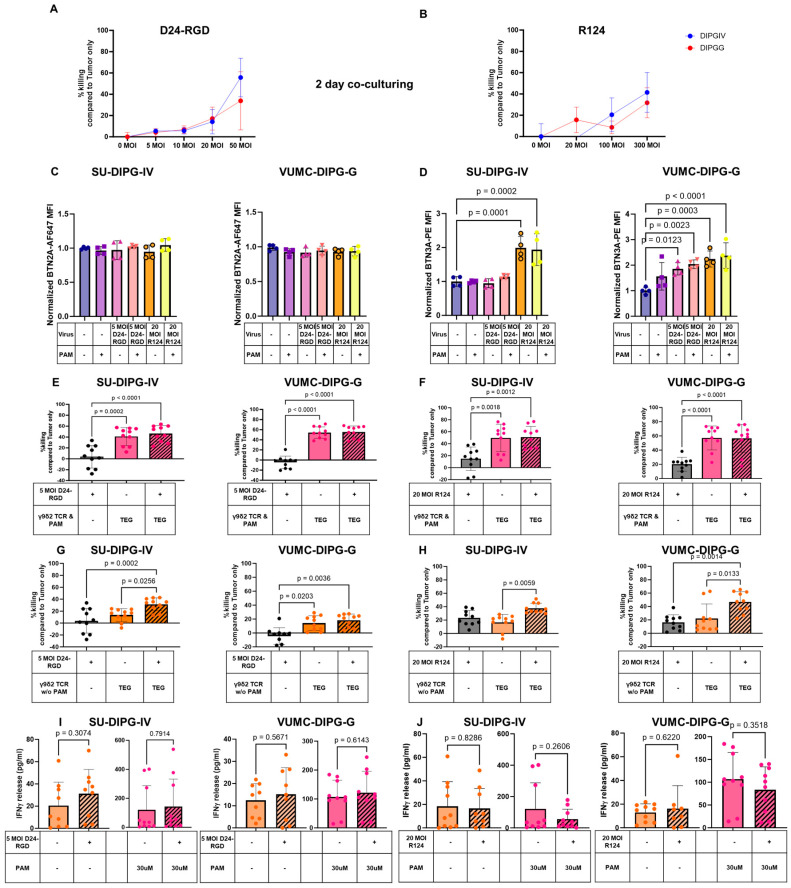
Combining oncolytic viruses with TEGs enhanced specific killing of patient-derived diffuse midline glioma cell cultures in a cell-specific manner and BTN2A1 and BTN3A-independent manner in the absence of PAM. Killing percentage of DMG cell SU-DIPG-IV and VUMC-DIPG-G 2 days post-infection with 5, 10, 20, and 50 MOI of D24-RGD (**A**) and with 20, 100, and 300 MOI of R124 (**B**), data are normalized on the number of alive tumor cells without any treatment (tumor only). BTN2A1 (**C**) and BTN3A (**D**) expression as normalized MFI vs. untreated SU-DIPG-IV and VUMC-DIPG-G, respectively, conditions tested included SU-DIPGIV and VUMC-DIPG-G infected with D24-RGD or R124 and in the presence or absence of 30 μM PAM for 48 h (**C**,**D**). The 2-day co-culture killing assays comparing the killing of DMGs infected with D24-RGD (**E**,**G**) or infected with R124 (**F**,**H**) in combination with TEG cells in the presence of 30 μM PAM (**E**,**F**) as well as in the absence of PAM (**G**,**H**). IFN-γ release fold change of co-cultures of TEG cells with SU-DIPG-IV and VUMC-DIPG-G infected with 5 MOI of D24-RGD (**I**) or 20 MOI R124 (**J**) in the absence of PAM. BTN2A1 and BTN3A experiments are represented as mean ± SD from n = 2 independent experiments of the normalized MFI to the untreated control. Statistics are assessed with a one-way ANOVA and corrected for multiple comparisons with the Tukey test. Data for killing assays from multiple independent experiments (n = 2) for SU-DIPG-IV and VUMC-DIPG-G for (**A**,**B**) while (n = 4) for (**E**–**H**) represented as dot points (biological replicates) and error bar as ± SD. Statistics were assessed with the one-way ANOVA and corrected for multiple comparisons with the Tukey test. Data for IFN-γ from n = 4 independent experiments are represented as (pg/mL) in dot points (biological replicates) and the error bar as ± SD. Statistics are assessed with the Student two-tailed *t*-test.

## Data Availability

Data is contained within the article and Appendix A.

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
