# Peer review of "Unusual Partners: γδ-TCR-Based T Cell Therapy in Combination with Oncolytic Virus Treatment for Diffuse Midline Gliomas"

_ijms, 2025, doi:10.3390/ijms26052167_

Round 1

Reviewer 1 Report

Comments and Suggestions for Authors

This study focused on limitations of current therapies for DMGs, providing a possible rationale for exploring alternative or combinatory approaches using TEGs and OV. From my perspective, the current data are insufficient for publication, some issues need to be supplemented.

1.     Line 215: Why was the co-culture assay for VUMC-DIPG-G conducted with only 2 independent experiments, while SU-DIPG-IV was conducted with 3? If the VUMC-DIPG-G group included only 2 independent experiments, how was statistical significance determined? Please clarify.

2.     Will the author be able to add dots to mark each sample in the bar charts (like Figure 2C/2D)?

3.     The IFN-γ release should also be detected in the TEG+OV group with or without PAM, rather than only TEG group. This could further support the synergistic effect of this combination strategy.

4.     The author only demonstrated that the use of OV increased the expression of BTN3A. Will the use of TEG enhance the infection rate of OV (detection of Hexon or E1A protein is recommended)? or enhance the expression of coxsackievirus and adenovirus receptor (CAR) on tumor cells which mediate the adenovirus entry?

5.     While the results are presented, the underlying reasons for the loss of additive effect with pamidronate are not discussed, leaving an important aspect unexplained.

Author Response

Dear Reviewer,

Thank you very much for contributing your time for this brief report. Please find the detailed responses below and the corresponding revisions/corrections highlighted/in track changes in the re-submitted files.

Reviewer 1:

This study focused on limitations of current therapies for DMGs, providing a possible rationale for exploring alternative or combinatory approaches using TEGs and OV. From my perspective, the current data are insufficient for publication, some issues need to be supplemented.

  1. Line 215: Why was the co-culture assay for VUMC-DIPG-G conducted with only 2 independent experiments, while SU-DIPG-IV was conducted with 3? If the VUMC-DIPG-G group included only 2 independent experiments, how was statistical significance determined? Please clarify.
  2. We incorrectly indicated 2 independent experiments for VUMC-DIPG-G, while in reality we included 3 independent experiments. We used 2 replicates per independent experiment, thus comparing 6 biological points per condition. Statistics were assessed with one-way ANOVA and corrected for multiple comparisons with the Tukey test. For further clarification dots have been added for each figure representing biological replicates, as now indicated in lines 224-226.
  3. Will the author be able to add dots to mark each sample in the bar charts (like Figure 2C/2D)?
  4. As discussed in the previous comment dots have been added in all figures (Figure 1A-C and Figure 2A-J).
  5. The IFN-γ release should also be detected in the TEG+OV group with or without PAM, rather than only TEG group. This could further support the synergistic effect of this combination strategy.
  6. We thank the reviewer for his/her comment, and we have now added the IFN-γ release of the TEG+OV groups with PAM in Figure 2I&J. In addition, we now present IFN-γ as pg/mL instead of normalized values. However, in the presence of PAM no statistical significant increase in IFN-γ was detected. Moreover, other published in vitro investigations show that successful OV combination with immune cell populations is not always followed by an IFN-γ increase (DOI: 10.1016/j.crmeth.2024.100716). Therefore, we suggest an additive rather than a synergistic effect of this combination, as we further elaborate on in the result and discussion section (lines 269-274 & 332-341).
  7. The author only demonstrated that the use of OV increased the expression of BTN3A. Will the use of TEG enhance the infection rate of OV (detection of Hexon or E1A protein is recommended)? or enhance the expression of coxsackievirus and adenovirus receptor (CAR) on tumor cells which mediate the adenovirus entry?
  8. We thank the reviewer for the suggestion, as this is an important point Marta Alonso et al. already demonstrated that differences in expression levels of Delta24-RGD entry receptors did not affect the OVs infection rate of diffuse midline- and high grade glioma cells (https://doi.org/10.1038/s41467-019-10043-0). In addition, we previously investigated the effect of D24-RGD and R124 infection in 14 Pediatric brain tumor cell cultures, and found that none of the known entry receptors of these OVs correlated with oncolysis, although their oncolytic efficacy was related to other factors unique per virus (DOI: 10.1016/j.omton.2024.200804). Therefore, we only stained virus specific proteins (Hexon for D24-RGD and σ3 for R124) to compare the infection rate in DMG cultures in the presence/absence of PAM, as suggested by the reviewer. In addition, we used Jurkat cells expressing γ9δ2 TCR receptor to investigate tumor-T cell interaction. In the figures included below, the presence of PAM reduced the percentage of D24-RGD or R124 infected DMG cells (Figure 1 and Figure 2 respectively). We only observed an increase of this infection in DIPG-G R124-infected cells (right panel, Figure 2), but this might also be attributed to the additional viral particles released by the R124-infected Jurkat cells (Figure 3). This would be in line with a publication by Alain et al. who demonstrated the ability of Reoviruses to infect lymphoid/leukemic cells and primary lymphoid neoplasias and CLL, (https://doi.org/10.1182/blood-2002-02-0503 ), while sparing the normal lymphocytes or hematopoietic stem/progenitor cells. It’s important to note that Jurkat cells were used to observe Tumor- T cell interaction without inducing tumor death, which we deemed to be important for this set-up.

D24-RGD

Figure 1: Percentage of anti-Hexon-AF488+ DMG cells in different conditions after 48h of D24-RGD infection and co-culture with γ9δ2 TCR expressing Jurkat cells. Conditions: 5 or 20 MOI D24RGD, 5 or 20 MOI D24RGD with 1:1 ratio of DMG : Jurkat cells, 5 or 20 MOI D24RGD with 30uM PAM, 5 or 20 MOI D24RGD with 1:1 ratio of DMG : Jurkat cells with 30uM PAM. The experiment set-up used was identical to the co-cultures described in the manuscript. The experiments were repeated two independent times and statistics were assed with one-way ANOVA and corrected for multiple comparisons with Tukey test.

R124

Figure 2: Percentage of anti-σ3 -AF488+ DMG cells in different conditions after 48h of R124 infection and co-culture with γ9δ2 TCR expressing Jurkat cells. Conditions: 20 or 100 MOI R124, 20 or 100 MOI R124 with 1:1 ratio of DMG : Jurkat cells, 20 or 100 MOI R124with 30uM PAM, 20 or 100 MOI R124with 1:1 ratio of DMG : Jurkat cells with 30uM PAM. The experiment set-up used was the same as the co-cultures of the described in the manuscript. The experiments were repeated two independent times and statistics were assed with one-way ANOVA and corrected for multiple comparisons with Tukey test.

Jurkat

Figure 3: Percentage of anti-Hexon-AF488+ Jurkat cells in different co-culture conditions after 48h of D24-RGD infection (left) and anti-σ3 -AF488+ Jurkat cells in different co-culture conditions after 48h of R124 infection. The Jurkat cells investigated originated from the co-culture experiments with DMG cells mentioned above. The experiments were repeated two independent times and statistics were assed with one-way ANOVA and corrected for multiple comparisons with Tukey test.

In conclusion our results demonstrate that in the presence of PAM  the viral infection of DIPGIV and DIPGG is reduced in most of the observed cases. This could partially explain the lack of combination effect in the presence of PAM.

  1. While the results are presented, the underlying reasons for the loss of additive effect with pamidronate are not discussed, leaving an important aspect unexplained.
  2. We thank the reviewer for this suggestion and now included a section in the discussion to address this issue. Here we describe the underlying mechanisms for the loss of an additive effect in the presence of pamidronate. Our hypothesis mainly revolves around the metabolic effects PAM has demonstrated in other tumors by reducing Ras and the decrease of membrane motility (https://doi.org/10.1016/j.jhep.2005.09.022). This could affect the viral cycle of the Ovs, and particularly of R124 which heavily prefers tumor cells with an activated Ras pathway as mentioned in the introduction line (65-67). We have included a part in the discussion section describing the loss of the additive effect with pamidronate line (332-341).

Reviewer 2 Report

Comments and Suggestions for Authors

The authors combined engineered T-cell therapy with oncolytic virus treatment to address pediatric diffuse midline glioma (DMG). This combination demonstrated synergistic effects in effectively targeting and killing DMG cells. The research was well-designed, and the conclusions were supported by the results.

However, the following recommendations are suggested for improvement:

  1. Have the authors considered assessing BTN3A and BTN2A1 expression using Western blotting?
  2. The instructions provided lack sufficient detail to fully inform the audience.

Author Response

Reviewer 2:

The authors combined engineered T-cell therapy with oncolytic virus treatment to address pediatric diffuse midline glioma (DMG). This combination demonstrated synergistic effects in effectively targeting and killing DMG cells. The research was well-designed, and the conclusions were supported by the results.

However, the following recommendations are suggested for improvement:

1.Have the authors considered assessing BTN3A and BTN2A1 expression using Western blotting?

  1. We thank the reviewer for his/her suggestion to use Western blotting for the detection of BTN3A and BTN2A1. However, as these proteins act as receptors for γδ-TCR, their presence in the tumor membrane is critical for the tumor-TEG engagement (https://doi.org/10.1016/j.celrep.2021.109359). Therefore, we considered staining and quantification of BTN3A and BTN2A1 through FACS more appropriate to assess their expression for the evaluation of TEG activity, as demonstrated in Figure 1A and Figure 2C,D.
  2. The instructions provided lack sufficient detail to fully inform the audience.
  3. We have appropriately increased the details about BTN3A and BTN2A1 expression and changes after OV infection. In detail, we expanded the method section about the BTN2A1 antibody used for FACS (lines 186-194), while highlighting the investigation of membranous BTN3A and BTN2A1 expression for TEG engagement throughout the result section (lines 207-209).

Reviewer 3 Report

Comments and Suggestions for Authors

Comments for the Authors

ijms-3407226

Unusual partners: γδ-TCR based T cell therapy in combination with Oncolytic virus treatment for Diffuse Midline Gliomas

In the manuscript, Vazaios et al. present the potential of γδ-TCR based T cell therapy associated with the presence of oncolytic viruses (adenovirus Δ24-RGD or reovirus R124), subsequently promoting the efficacy of the therapy (killing of pediatric diffuse midline gliomas).

This manuscript is well written and the main theme is well expressed. I only have a couple of comments:

1. Experimental results demonstrate that different oncolytic viruses may lead to different therapeutic outcomes. Although the authors have briefly mentioned the difference in tumor tropisms across oncolytic viruses, I recommend that the authors illustrate in more detail how mechanistically oncolytic viruses can aid in killing tumor cells and provide their perspectives about the potential of using oncolytic viruses towards immunotherapies (e.g., what are criteria to choose proper viruses for therapies and are viruses tumor-specific or can be used for killing a wide range of tumors?).

2. I suggest the author replace the label of y axes (BTN2A-AF647 and BTN3A-PE) in Figure 2C and 2D by the same terms used in the main text to keep consistency. 

3. Did the authors repeat the experiments on animals? How much can the results observed in cellular models represent those observed in living animals?

Author Response

Dear Reviewer, thank you very much for taking the time to review this brief report. Please find the detailed responses below and the corresponding revisions/corrections highlighted/in track changes in the re-submitted files.

Reviewer 3:

  1. Experimental results demonstrate that different oncolytic viruses may lead to different therapeutic outcomes. Although the authors have briefly mentioned the difference in tumor tropisms across oncolytic viruses, I recommend that the authors illustrate in more detail how mechanistically oncolytic viruses can aid in killing tumor cells and provide their perspectives about the potential of using oncolytic viruses towards immunotherapies (e.g., what are criteria to choose proper viruses for therapies and are viruses tumor-specific or can be used for killing a wide range of tumors?).
  2. We thank the reviewer for his/her suggestion and now included a more detailed description of the criteria of a desirable OV, the factors affecting their tropism and efficacy and their use as immune modulators in the discussion (lines 353- 367). More specifically, we mention that OVs require some characteristics to be used for cancer therapy, as discussed in the review “Oncolytic Virotherapy: A New Paradigm in Cancer Immunotherapy” by Simona Volovat et. al (https://doi.org/10.3390/ijms25021180). Though specific OVs have higher tropism for specific tumors, their tropism toward different tumor types is not limited by type or entity. Tumor-specific intrinsic factors seem to be the deciding factor of successful OV action (DOI: 10.1016/j.omton.2024.200804, https://doi.org/10.3390/vaccines9101166). In addition, OVs are able to act as immune modulators as observed by their effects on immune stimulation and increased infiltration by immune cells as summarized from preclinical and clinical studies in our review (DOI: 10.3390/ijms25095007 ).
  3. I suggest the author replace the label of y axes (BTN2A-AF647 and BTN3A-PE) in Figure 2C and 2D by the same terms used in the main text to keep consistency. 
  4. Thank you for notifying us, we have adjusted this accordingly.
  5. Did the authors repeat the experiments on animals? How much can the results observed in cellular models represent those observed in living animals?
  6. Thank you for pointing this out. We used patient-derived cell cultures as a proof of concept for investigating the direct effects of OVs, tumors and TEGs as similar experimental designs have also been used before to investigate similar interactions (DOI: 10.1016/j.cell.2018.07.009, https://doi.org/10.1038/s41587-022-01397-w ). However, we cannot respond with certainty on how representative the results we observed would be in living animals due to many reasons. First, OVs demonstrate limited species tropism and therefore replicate very poorly in other species. Second, human adenoviruses would require patient derived xenografts (PDXs) to investigate their infective and oncolytic capabilities.  However, PDXs can only be established in immunodeficient mice that lack a functional immune system, removing the effect of the immunosuppressive tumor microenvironment (TME), a limitation also present in in vitro models. Thus, for the investigation of Delta24-RGD the use of more uncommon animal platforms would be required such as immunocompetent hamsters or humanized mouse models (https://doi.org/10.1158/1078-0432.CCR-16-0477). In vitro models using patient-derived material have been used to investigate and discover novel therapies that could translate to patients (https://doi.org/10.1101/2020.12.29.424674, https://doi.org/10.1016/j.ccell.2023.03.007 )in an attempt to overcome the limited translational efficacy from bench to beside usually observed in animal experimentation (DOI: 10.1186/s12967-018-1678-1). We included a small paragraph discussing this topic in the discussion section putting our results in perspective (lines 368-380).

Round 2

Reviewer 1 Report

Comments and Suggestions for Authors

The author revised this manuscript correctly, I have no further issue

Reviewer 3 Report

Comments and Suggestions for Authors

I think authors' responses to my comments. I do not have further questions about this current version of the manuscript.